# Video Object Segmentation with Adaptive Feature Bank and Uncertain-Region Refinement

**Yongqing Liang, Xin Li**,* **Navid Jafari**
Louisiana State University
{ylian16, xinli, njafari}@lsu.edu

**Qin Chen**
Northeastern University
q.chen@northeastern.edu

## Abstract

This paper presents a new matching-based framework for semi-supervised video object segmentation (VOS). Recently, state-of-the-art VOS performance has been achieved by matching-based algorithms, in which feature banks are created to store features for region matching and classification. However, how to effectively organize information in the continuously growing feature bank remains under-explored, and this leads to an inefficient design of the bank. We introduced an adaptive feature bank update scheme to dynamically absorb new features and discard obsolete features. We also designed a new confidence loss and a fine-grained segmentation module to enhance the segmentation accuracy in uncertain regions. On public benchmarks, our algorithm outperforms existing state-of-the-arts.

## 1   Introduction

Video object segmentation (VOS) is a fundamental step in many video processing tasks, like video editing and video inpainting. In the semi-supervised setting, the first frame annotation is given, which depicts the objects of interest of the video sequence. The goal is to segment mask of that object in the subsequent frames. Many deep learning based methods have been proposed to solve this problem in recent years. When people tackle the semi-supervised VOS task, the segmentation performance is affected by two main steps: (1) distinguish the object regions from the background, (2) segment the object boundary clearly.

A key question in VOS is how to learn the cues of target objects. We divide recent works into two categories, *implicit learning* and *explicit learning*. Conventional *implicit approaches* include detection-based and propagation-based methods [3, 27, 10, 32, 2, 13, 23]. They often adopt the fully convolutional network (FCN) [20] pipeline to learn object features by the network weights implicitly; then, before segmenting a new video, these methods often need an online learning to fine-tune their weights to learn new object cues from the video. *Explicit approaches* learn object appearance explicitly. They often formulate the segmentation as pixel-wise classification in a learnt embedding space [31, 4, 24, 11, 33, 18, 12, 25, 17]. These approaches first construct an embedding space to memorize the object appearance, then segment the subsequent frames by computing similarity. Therefore, they are also called matching-based methods. Recently, matching-based methods achieve the state-of-the-art results in the VOS benchmark.

A fundamental issue in matching-based VOS segmentation is how to effectively exploit previous frames' information to segment the new frame. Since the memory size is limited, it is not possible and unnecessary to memorize information from all the previous frames. Most methods [24, 33, 31, 18, 25] only utilize the first and the latest frame or uniformly sample key frames. However, when the given video becomes longer, these methods often either miss sampling on some key-frames or encounter

out-of-memory crash. To tackle this problem, we propose an *adaptive feature bank* (AFB) to organize the target object features. This adaptive feature bank absorbs new features by weighted averaging and discards obsolete features according to the least frequently used (LFU) index. As results, our model could memorize the characteristics of multi objects and segment them simultaneously in long videos under a low memory consumption.

Besides identifying the target object, clearly segmenting object boundary is also critical to VOS performance: (1) People are often sensitive to boundary segmentation. (2) When estimated masks on some boundary regions are ambiguous and hard to classify, their misclassificaton is easily accumulated in video. However, most recent VOS methods follow an encoder-decoder mode to estimate the object masks, the boundary of the object mask becomes vague when it is iteratively upscaled from a lower resolution. Therefore, we propose an *uncertain-region refinement* (URR) scheme to improve the segmentation quality. It includes a novel classification confidence loss to estimate the ambiguity of segmentation, and a local fine-grained segmentation to refine the ambiguous regions.

Our **main contributions** are three-folded: (1) We proposed an adaptive and efficient feature bank to maintain most useful information for video object segmentation. (2) We introduced a confidence loss to estimate the ambiguity of the segmentation results. We also designed a local fine-grained segmentation module to refine these ambiguous regions. (3) We demonstrated the effectiveness of our method on segmenting long videos, which are often seen in practical applications.

## 2 Related Work

Recent video object segmentation works can be divided into two categories: *implicit learning* and *explicit learning*. The *implicit learning* approaches include detection-based methods [3, 23] which segment the object mask without using temporal information, and propagation-based methods [27, 10, 32, 2, 13, 9] which use masks computed in previously frames to infer masks in the current frame. These methods often adopt a fully convolutional network (FCN) structure to learn object appearance by network weights implicitly; so they often require an online learning to adapt to new objects in the test video.

The *explicit learning* methods first construct an embedding space to memorize the object appearance, then classify each pixel's label using their similarity. Thus, the explicit learning is also called *matching-based* method. A key issue in matching-based VOS segmentation is how to build the embedding space. DMM [36] only uses the first frame's information. RGMP [24], FEELVOS [31], RANet [33] and AGSS [18] store information from the first and the latest frames. VideoMatch [11] and WaterNet [17] store information from several latest frames using a slide window. STM [25] stores features every $T$ frames ($T = 5$ in their experiments). However, when the video to segment is long, these static strategies could encounter out-of-memory crashes or miss sampling key-frames. Our proposed *adaptive feature bank* (AFB) is a first non-uniform frame-sampling strategy in VOS that can more flexibly and dynamically manage objects' key features in videos. AFB performs dynamic feature merging and removal, and can handle videos with any length effectively.

Recent image segmentation techniques introduce fine-grained modules to improve local accuracy. ShapeMask [15] revise the decoder from FCN to refine the segmentation. PointRend [14] defines uncertainty on a binary mask, and does a one-pass detection and refinement on uncertain regions. We proposed an *uncertainty-region refinement* (URR) strategy to perform boundary refinement in video segmentation. URR includes (1) a more general multi-object uncertainty score for estimated masks, (2) a novel confidence loss to generate cleaner masks, and (3) a non-local mechanism to more reliably refine uncertain regions.

## 3 Approach

The overview of our framework is illustrated in Fig. 1. First, as shown in blue region, we use a basic pipeline of matching-based segmentation (Sec. 3.1) to generate initial segmentation masks. In Sec. 3.2, we propose an adaptive feature bank module to dynamically organize the past frame information. In Sec. 3.3, given the initial segmentation, we design a confidence loss to estimate the ambiguity of misclassification, and a fine-grained module to classify the uncertain regions. These two components are marked in the red region in Fig. 1.

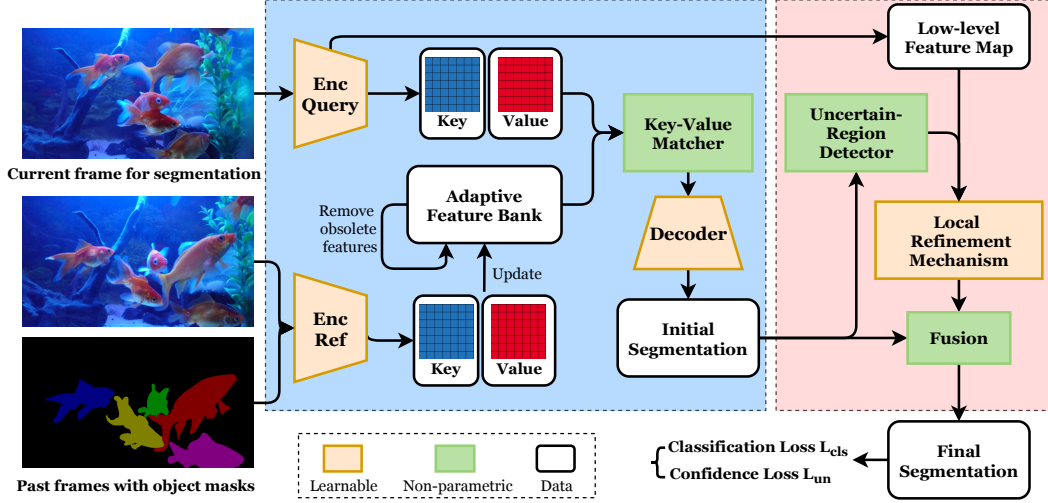

Figure 1: Algorithm Overview. A typical matching-based module (in light blue) is used to estimate the initial segmentation. An adaptive feature bank is proposed to organize the feature space. A novel uncertain-region refinement mechanism (in pink) is designed for fine-grained segmentation.

## 3.1 Matching-based Segmentation

Given an evaluation video, we encode the first frame and its groundtruth annotation to build a feature bank. Then, we use the feature bank to match and segment target objects starting from the second frame. The decoder takes the matching results to estimate the frame's object masks.

**Encoders.** A *query encoder* is designed to encode the current frame, named query frame, to its feature map for segmentation. We use the ResNet-50 [8] as backbone and takes the output of the layer-3 as feature map $Q \in \mathbb{R}^{(H/8) \times (W/8) \times 1024}$, where $H$ and $W$ are the height and width.

For segmenting the $t$th frame, we treat the past frames from 1 to $t-1$ as reference frames. A *reference encoder* is designed for memorizing the characteristics of the target objects. Suppose there are $L$ objects of interest, we encode the reference frame object by object and output $L$ feature maps $\hat{P}_i$, $i \in [1, L]$. The reference encoder is a modification of the original ResNet-50. For each object $i$, it takes both the reference frame and its corresponding mask as inputs, then extracts object-level feature map, $\hat{P}_i \in \mathbb{R}^{(H/8) \times (W/8) \times 1024}$. Combining the object-level feature maps together, we obtain the feature maps of the reference frame at index $j$, $P_j = \{\hat{P}_1, \hat{P}_2, \cdots, \hat{P}_L\}$, where $j \in [1, t-1]$.

**Feature map embedding.** Traditional matching-based methods directly compare the feature maps of the query feature map and the reference feature maps. Although such design is good for classification, they are lack of semantic information to estimate the object masks. Inspired by STM [25], we utilize the similar feature map embedding module. The feature maps are encoded into two embedding spaces, named *key* $k$ and *value* $v$ by two convolutional modules. Specifically, we match the feature maps by their keys $k$, while allow their values $v$ being different in order to preserve as much semantic information as possible. The feature bank stores pairs of keys and values of the past frames. In the next section, we compare the feature maps of the current frame with the feature bank to estimate the object masks. The details of maintaining the feature bank are elaborated in Section 3.2.

**Matcher.** A query frame is encoded into pairs of key $k^Q$ and value $v^Q$ through the query encoder and feature map embedding. We maintain $L$ feature banks $FB_i, i \in [1, L]$ from the past frames for each object $i$. The similarity between the query frame and the feature banks is calculated object by object. For each point $p$ in the query frame, we use a weighted summation to retrieve the closest value $\hat{v}_i(p)$ in the $i$th object feature bank,

$$\hat{v}_i(p) = \sum_{(k^{FB}, v^{FB}) \in FB_i} g(k^Q(p), k^{FB}) v^{FB}, \qquad (1)$$

where $i \in [1, L]$, and $g$ is the softmax function $g(k^Q(p), k^{FB}) = \frac{exp(k^Q(p) \bullet k^{FB})}{\sum_{\forall j} exp(k^Q(p) \bullet k^{FB})}$, where $\bullet$ represents the dot production between two vectors. We concatenate the query value map with its most similar retrieval value map as $y_i = [v^Q, \hat{v}_i], \quad i \in [1, L]$, where $y_i$ is the matching results between the query frame and the feature banks for the object $i$.

**Decoder.** The decoder takes the output of the matcher $y_i, i \in [1, L]$ to estimate the object mask independently, where $y_i$ depicts the semantic information for the object $i$. We follow the refinement module used in [24, 18, 25] that gradually upscales the feature map by a set of residual convolutional blocks. At each stage, the refinement module takes both the output of the previous stage and a feature map from the query encoder at corresponding scale through skip connections. After the decoder module, we obtain the initial object masks $M_i$ for each object $i$. We minimize the cross entropy loss $\mathcal{L}_{cls}$ between the object masks and the groundtruth labels $C$. The $\mathcal{L}_{cls}$ is averaged across all pixels $p$,

$$\mathcal{L}_{cls}(M, C) = -\frac{1}{|p|} \sum_p \left[ log\left(\frac{exp(M_c)}{\sum_i exp(M_i)}\right) \right]_p. \tag{2}$$

## 3.2 Adaptive Feature Bank

We build a feature bank to store features of each object and classify new pixels/regions in the current frame $t$. Storing features from all the previous frames $\{1, \ldots, t-1\}$ is impossible, because it will make the bank prohibitively big (as the length of the video clip grows) and make the query slow. Recent approaches either store the features from every few frames or from the several latest frames. For example, STM [25] uniformly stores every one of $K = 5$ frames in feature bank; and on an NVIDIA 1080Ti card with 11GB Memory, this can only handle a single-object video at most 350+ frames. Practical videos are often longer (e.g., average YouTube video has about 12 minutes or 22K frames); and for a 10-min video, STM needs to set $K = 300$, which will probably miss many important frames and information. Hence, we propose an *adaptive feature bank* (AFB) to more effectively manage object's key features. AFB contains two operations: absorbing new features and removing obsolete ones.

**Absorbing new features.** In video segmentation, although features from most recent frames are often more important, earlier frames may contain useful information. Therefore, rather than simply ignoring those earlier frames, we keep earlier features and organize all the features by a weighted averaging (which has been shown effective for finding optimal data representations such as Neural Gas [7]). Fig. 2 illustrates how the adaptive feature bank absorbs new features. Existing features and new features are marked in blue and red, separately. When a new feature is extracted, if it is close enough to some existing one, then merge them. Such a merge avoids storing redundant information and helps the memory efficiency. Meanwhile, it allows a flexible update on the stored features according to the object's changing appearance.

At the beginning of the video segmentation, the feature bank is initialized by the features of the first frame. We build an independent feature bank for each object. Since the object-level feature banks are maintained separately, we omit the object symbol here to simplify the formulae. Suppose we have estimated the object mask of the $(t-1)$th frame, then the $(t-1)$th frame and the estimated mask are encoded into features $(k_{t-1}^P, v_{t-1}^P)$ through the reference encoder and the embedding module. For each new feature $a(i) = (k_{t-1}^P(i), v_{t-1}^P(i))$ and old features that store in the feature bank $b(j) = (k^{FB}(j), v^{FB}(j)) \in FB$, we employ an inner product as the similarity function,

$$h(a(i), b(j)) = \frac{k_{t-1}^P(i) \bullet k^{FB}(j)}{\|k_{t-1}^P(i)\| \|k^{FB}(j)\|}. \tag{3}$$

For each new feature $a(i)$, we select the most similar feature $b(j')$ from the feature bank that $H(a(i)) = max_{\forall b(j) \in FB} h(a(i), b(j))$. If $H(a(i))$ is large enough, since these two features are similar, we will merge the new one to the feature bank. Specifically, when $H(a(i)) > \epsilon_h$,

$$\begin{aligned} k^{FB}(j') &= (1 - \lambda_p)k^{FB}(j') + \lambda_p k_{t-1}^P(i), \\ v^{FB}(j') &= (1 - \lambda_p)v^{FB}(j') + \lambda_p v_{t-1}^P(i), \end{aligned} \tag{4}$$

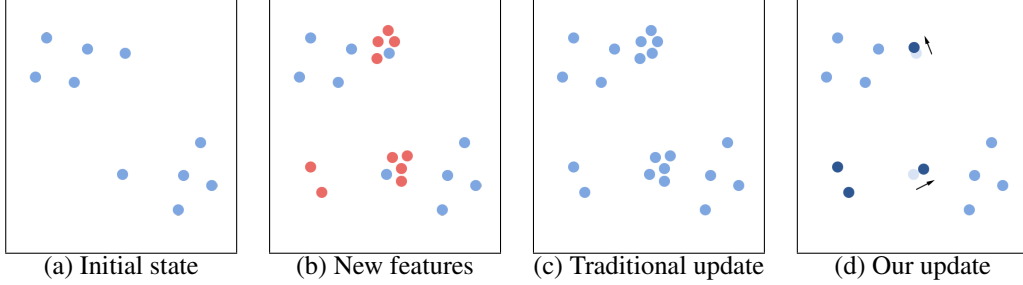

| (a) Initial state | (b) New features | (c) Traditional update | (d) Our update |

Figure 2: An illustration of the Adaptive Feature Bank. (a) shows an initial state of a feature bank, where blue points are the key features. In (b), new extracted features to add are marked in red. In the traditional feature bank (c), features are directly added and it produces redundant features. In our design, as (d) shows, some features are alternated (in dark blue) to absorb those similar new features. The black arrows show the directions of the moving average.

where $\epsilon_h = 0.95$ controls the merging rate, and $\lambda_p = 0.1$ controls the impact of the moving averaging. Otherwise, for all $H(a(i)) \leq \epsilon_h$, because the new features are so distinct from all the existing ones, we append the new features to the feature bank,

$$
\begin{aligned}
k^{FB} &= k^{FB} \cup k_{t-1}^P(i), \\
v^{FB} &= v^{FB} \cup v_{t-1}^P(i).
\end{aligned}
\tag{5}
$$

From our experiments, we find about $90\%$ of new features that satisfy merging operation and we only need to add the rest $10\%$ each time.

**Removing obsolete features.** Though the above *updating* strategy reliefs memory pressure significantly (e.g., $90\%$ less memory consumption), feature bank sizes still gradually expand with the growth of frame number. Similar to the cache replacement policy, we measure which old features are least likely to be useful and may be eliminated. We build a measurement using the least-frequently used (LFU) index.

Each time when we use the feature bank to match the query frame in Eqn. 1, if the similarity function $g$ is greater than a threshold $\epsilon_l = 10^{-4}$, we increase the count of this feature. Specifically, for $\forall (k^{FB}(j), v^{FB}(j)) \in FB$, the LFU index is counted by

$$
cnt(j) := cnt(j) + log\Big( \sum_{\forall (k^Q(i), v^Q(i))} sgn\big(g(k^Q(i), k^{FB}(j)) > \epsilon_l\big) + 1\Big),
$$
$$
LFU(j) = \frac{cnt(j)}{l(j)},
\tag{6}
$$

where $l(j)$ is the time span that the feature stays in the feature bank and the $log$ function is used to smooth the LFU index. In practice, when the size of the feature bank is about to exceed the predefined budget, we remove the features with the least LFU index until the size of the feature bank is below the budget. The LFU index counting and feature removing procedure are very efficient. This adaptive feature bank scheme can be generalized to other matching-based video processing methods to maintain the bank size, making it suitable to handle videos of arbitrary length.

### 3.3 Uncertain-region Refinement

In the decoding stage, the object masks are computed from the upscaled low-resolution images. Therefore, the object boundaries of such estimated masks are often ambiguous. The classification accuracy of the boundary regions, however, is critical to the segmentation results. Hence, we propose a new scheme, named *uncertain-region refinement* (URR), to tackle boundary and other uncertain regions. It includes a new loss to evaluate such uncertainty and a novel local refinement mechanism to adjust fine-grained segmentation.

**Confidence loss.** After decoding and a softmax normalization, we have a set of initial segmentations $M_i$ for each object $i$, $i \in [1, L]$. The object mask $M_i$ represents the likelihood of each pixel $p$

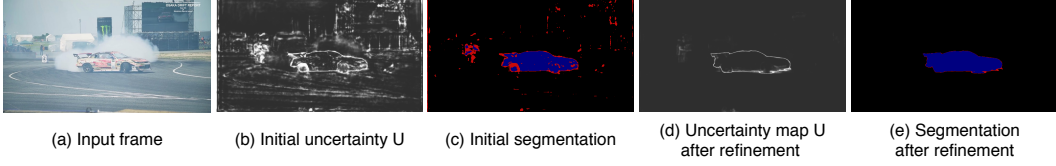

| (a) Input frame | (b) Initial uncertainty U | (c) Initial segmentation | (d) Uncertainty map U after refinement | (e) Segmentation after refinement |

Figure 3: An illustration of the Uncertain-regions Refinement. (a) is an input frame. (b) is the initial uncertainty map $U$, where brightness indicates the uncertainty. (c) is the initial segmentation, where blue regions are the object masks and red regions highlights regions whose uncertainty score $U > 0.7$. In (d), the uncertain-regions refinement helps classify ambiguous areas. (e) is the final segmentation.

belonging to the object $i$, where the value range of $M_i$ is in $[0, 1]$, and $\sum_{i=1}^{L} M_i(p) = 1$. In other words, for each pixel $p$, there are $L$ values $M_i(p)$ in $[0, 1]$, indicating the likelihood of $p$ being one of these $L$ objects. We can simply pick the index $i$ of the largest value $M_i(p)$ as $p$'s label.

More adaptively, we define a *pixel-wise uncertainty map* $U$ to measure classification ambiguity on each pixel, using the ratio of the largest likelihood value $\hat{M}^1$ to the second largest value $\hat{M}^2$,

$$U = exp(1 - \frac{\hat{M}^1}{\hat{M}^2}), \tag{7}$$

where $\frac{\hat{M}^1}{\hat{M}^2} \in [1, +\infty)$. The uncertainty map $U$ is in $(0, 1]$, where smaller value means more confidence. The confidence loss $\mathcal{L}_{conf}$ of a set of object masks is defined as

$$\mathcal{L}_{conf} = \|U\|_2. \tag{8}$$

During the training stage, our framework is optimized using the following loss function,

$$\mathcal{L} = \mathcal{L}_{cls} + \lambda_u \mathcal{L}_{conf}, \tag{9}$$

where the $\lambda_u = 0.5$ is a weight scalar. $\mathcal{L}_{cls}$ (Eqn. 2) is the cross entropy loss for pixel-wise classification. $\mathcal{L}_{conf}$ is designed for minimizing the ambiguities of the estimated masks, i.e., pushing each object mask towards a $0/1$ map.

**Local refinement mechanism.** We propose a novel local refinement mechanism to refine the ambiguous regions. Experimentally, given two neighbor points in the spatial space, if they belong to the same object, their features are usually close. The main intuition is that we use the pixels which have high confidence in classification to refine the other uncertain points in its neighborhood. Specifically, for each uncertain pixel $p$, we compose its local reference features $y(p) = \{y_i(p) | i \in [1, L]\}$ from $p$'s neighborhood, where $L$ is the number of target objects. If the local feature $r(p)$ of $p$ is close to the $y_i(p)$, we say pixel $p$ is likely to be classified as the object $i$. The local reference feature $y_i(p)$ is computed by weighted average in a small neighborhood $\mathcal{N}(p)$,

$$y_i(p) = \frac{1}{\sum_{q \in \mathcal{N}(p)} M_i(q)} \sum_{q \in \mathcal{N}(p)} M_i(q) r(q), \tag{10}$$

where the weight $M_i$ is the object mask for the object $i$. Then, a residual network module $f_l$ is designed to learn to predict the local similarity. We assign a local refinement mask $e$ for each pixel $p$ by comparing the similarity between $r(p)$ and $y_i(p)$,

$$e_i(p) = c_i(p) f_l\big(r(p), y_i(p)\big), \tag{11}$$

where $c_i(p) = max_{q \in \mathcal{N}(p)} M_i(q)$. $c_i$ are confidence scores for adjusting the impact of the local refinement mask.

Finally, we obtain the final segmentation $S_i$ for each object $i$ by adding the local refinement mask $e_i$ to the initial object mask $M_i$,

$$S_i(p) = M_i(p) + U(p) e_i(p). \tag{12}$$

Fig. 3 shows the effectiveness of the proposed *uncertain-region refinement* (URR). The initial segmentation (marked in blue) is ambiguous, where some areas are lack of confidence (marked in red). As Fig. 3 (d)(e) shown, when our model is trained with the $\mathcal{L}_{conf}$, the uncertain-region refinement improves the segmentation quality.

# 4 Training Details

Our model is first pretrained on simulation videos which is generated from static image datasets. Then, for different benchmarks, our model is further trained on their training videos.

**Pretraining on image datasets.** Since we don't introduce any temporal smoothness assumptions, the learnable modules in our model does not require long videos for training. Pretraining on image datasets is widely used in VOS methods [27, 25], we simulate training videos by static image datasets [5, 29, 16, 19, 6] ($136, 032$ images in total). A synthetic video clip has 1 first frame and 5 subsequent frames, which are generated from the same image by data augmentation (random affine, color, flip, resize, and crop). We use the first frame to initialize the feature bank and the rest 5 frames consist of a mini-batch to train our framework by minimizing the loss function $\mathcal{L}$ in Eqn. 9.

**Main training on the benchmark datasets.** Similar to the pretrianing routine, we randomly select 6 frames per training video as a training sample and apply data augmentation on those frames. The input frames are randomly resized and cropped into $400 \times 400$px for all training. For each training sample, we randomly select at most 3 objects for training. We minimize our loss using AdamW [21] optimizer ($\beta = (0.9, 0.999)$, $eps = 10^{-8}$, and the weight decay is 0.01). The initial learning rate is $10^{-5}$ for pretraining and $4 \times 10^{-6}$ for main training. Note that we directly use the network output without post-processing or video-by-video online training.

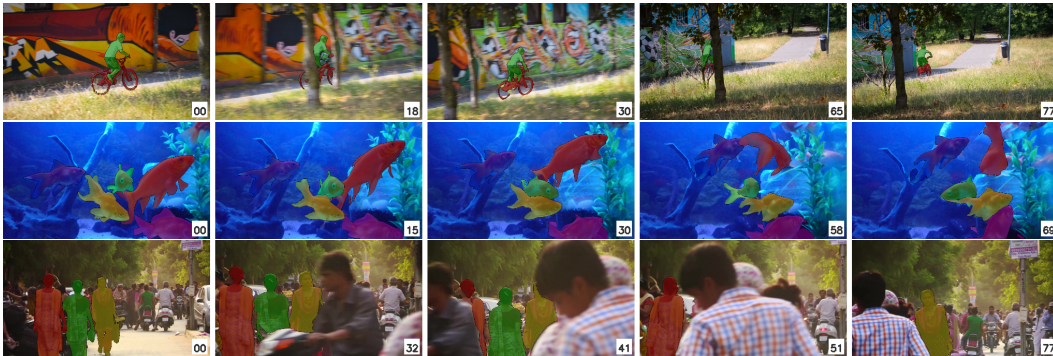

Figure 4: The qualitative results of our framework on DAVIS dataset. Frames of challenging moments (with occlusions or deformations) are shown. The target objects are marked in red, green, yellow, etc.

# 5 Experiments

## 5.1 Datasets and Evaluation Metrics

We evaluated our model (AFB-URR) on DAVIS17 [28] and YouTube-VOS18 [35], two large-scale VOS benchmarks with multiple objects. DAVIS17 contains 60 training videos and 30 validation videos. YouTube-VOS18 (YV) contains $3, 471$ training videos and 474 videos for validation. We implemented our framework in PyTorch [26] and conducted experiments on a single NVIDIA 1080Ti GPU. Qualitative results of our framework on DAVIS17 dataset are shown in Fig. 4. More qualitative comparisons are reported in the supplementary file.

We adopted the evaluation metrics from the DAVIS benchmark. The region accuracy $\mathcal{J}$ calculates the intersection-over-union (IoU) of the estimated masks and the groundtruth masks. The boundary accuracy $\mathcal{F}$ measures the accuracy of boundaries, via bipartite matching between the boundary pixels.

## 5.2 Comparison with the State-of-the-art

**Results on DAVIS benchmarks.** We compared our approach with recent implicit and explicit learning methods. Table 1 reports three accuracy scores [28] in percentages: the mean (M) is the average value, recall (R) measures the fraction of sequences scoring higher than a threshold $\tau = 0.5$, and decay (D) measures how the performance changes over time. Our method significantly

Table 1: The quantitative evaluation on the validation set of the DAVIS17 benchmark [28] in percentages. +YV indicates the use of YouTube-VOS for training. OL means it needs online learning.

| Methods | OL | $\mathcal{J}$ M | $\mathcal{J}$ R | $\mathcal{J}$ D | $\mathcal{F}$ M | $\mathcal{F}$ R | $\mathcal{F}$ D | $\mathcal{J}\&\mathcal{F}$ M |
|---|---|---|---|---|---|---|---|---|
| RANet [33] | | 63.2 | 73.7 | 18.6 | 68.2 | 78.8 | 19.7 | 65.7 |
| AGSS [18] | | 63.4 | - | - | 69.8 | - | - | 66.6 |
| RGMP [24] | | 64.8 | 74.1 | 18.9 | 68.6 | 77.7 | 19.6 | 66.7 |
| OSVOS$^S$ [23] | Yes | 64.7 | 74.2 | 15.1 | 71.3 | 80.7 | 18.5 | 68.0 |
| CINM [2] | Yes | 67.2 | 74.5 | 24.6 | 74.0 | 81.6 | 26.2 | 70.6 |
| A-GAME (+YV) [12] | | 68.5 | 78.4 | 14.0 | 73.6 | 83.4 | 15.8 | 71.0 |
| FEELVOS (+YV) [31] | | 69.1 | 79.1 | 17.5 | 74.0 | 83.8 | 20.1 | 71.5 |
| STM [25] | | 69.2 | - | - | 74.0 | - | - | 71.6 |
| **Ours** | | **73.0** | **85.3** | **13.8** | **76.1** | **87.0** | **15.5** | **74.6** |

Table 2: The quantitative evaluation on the validation set of the YouTube-VOS18 benchmark [35] in percentages. OL means it needs online learning.

| | MSK [27] | RGMP [24] | OnAVOS [32] | AGAME [12] | PreM [22] | S2S [34] | AGSS [18] | STM [25] | **Ours** |
|---|---|---|---|---|---|---|---|---|---|
| Need OL | Yes | | Yes | | Yes | Yes | | | |
| $\mathcal{J}$ seen | 59.9 | 59.5 | 60.1 | 66.9 | 71.4 | 71.0 | 71.3 | **79.7** | 78.8 |
| $\mathcal{J}$ unseen | 45.0 | - | 46.6 | 61.2 | 56.5 | 55.5 | 65.5 | 72.8 | **74.1** |
| $\mathcal{F}$ seen | 59.5 | 45.2 | 62.7 | - | 75.9 | 70.0 | 75.2 | **84.2** | 83.1 |
| $\mathcal{F}$ unseen | 47.9 | - | 51.4 | - | 63.7 | 61.2 | 73.1 | 80.9 | **82.6** |
| Overall | 53.1 | 53.8 | 55.2 | 66.0 | 66.9 | 64.4 | 71.3 | 79.4 | **79.6** |

outperforms existing methods: our $\mathcal{J}\&\mathcal{F}$ M score is 74.6 without any online fine-tune. Our model also has better runtime performance than the baseline STM [25]. On DAVIS17, with an NVIDIA 1080Ti, STM achieves 3.4fps with $\mathcal{J}\&\mathcal{F} = 71.6$, and ours achieves 4.0fps with $\mathcal{J}\&\mathcal{F} = 74.6$. We can also trade accuracy for better efficiency: if we limit the memory usage under $20\%$, it achieves 5.7fps with $\mathcal{J}\&\mathcal{F} = 71.7$.

**Results on YouTube-VOS benchmarks.** The validation set contains 474 first-frame-annotated videos. They include objects from 65 training categories and 26 unseen categories in training. Table 2 shows a comparison with previous state-of-the-art methods on the open evaluation server [35]. Our framework achieves the best overall score 79.6 because the adaptive feature bank improves the robustness and reliability for different scenarios. For those videos whose objects are already been seen in the training videos, STM's results are somewhat better than ours. The reason could be that their model and ours are evaluated on different memory budgets. STM [25] evaluated their work on an NVIDIA V100 GPU with 16GB memory, while we evaluated ours on a weaker machine (one NVIDIA 1080Ti GPU with 11GB memory). However, our framework is more robust in segmenting unseen objects, 74.1 in $\mathcal{J}$ and 82.6 in $\mathcal{F}$, compared with the STM's 72.8 in $\mathcal{J}$ and 80.9 in $\mathcal{F}$. Overall, our proposed model has great generalizability and achieves the state-of-the-art performance.

## 5.3 Segmentation of Long Videos

Table 3: The quantitative evaluation on the Long Videos in percentages.

| Methods | $\mathcal{J}$ M | $\mathcal{J}$ R | $\mathcal{J}$ D | $\mathcal{F}$ M | $\mathcal{F}$ R | $\mathcal{F}$ D | $\mathcal{J}\&\mathcal{F}$ M |
|---|---|---|---|---|---|---|---|
| RVOS [30] | 10.2 | 6.67 | 13.0 | 14.3 | 11.7 | **10.1** | 12.2 |
| A-GAME [12] | 50.0 | 58.3 | 39.6 | 50.7 | 58.3 | 45.2 | 50.3 |
| STM [25] | 79.1 | 88.3 | 11.6 | 79.5 | 90.0 | 15.4 | 79.3 |
| **Ours** | **82.7** | **91.7** | **11.5** | **83.8** | **91.7** | 13.9 | **83.3** |

The widely used benchmarks DAVIS17 (67 frames per video on average) and YouTube-VOS (132 frames per video on average) only contain trimmed short clips. To better evaluate the performance of our method in processing long videos in real-world tasks, we also conducted experiments on several long videos[2]. We randomly selected three videos from the Internet, that are longer than 1.5K frames and have their main objects continuously appearing. Each video has 20 uniformly sampled frames manually annotated for evaluation. Table 3 reports the experimental results of ours and three other state-of-the-art methods, ran on an NVIDIA 1080Ti (11GB Memory). We used these methods' released codes and their models pretrained on YouTube-VOS dataset. Note that STM [25] could only store at most 50 frames per video under 11GB GPU memory, so we set this parameter to 50.

RVOS [30] and A-GAME [12] achieved lower scores because they only use information from the first or the latest frame. They failed to segment the object of interest after 1K frames. STM has a $\mathcal{J}\&\mathcal{F}$ score of 79.3. Because the total frames that STM can store is fixed, when the video length becomes longer, STM has to increase the key frame interval and has a higher chance of missing important frames and information. In contrast, the proposed AFB mechanism can dynamically manage the key information from previous frames. We achieved the best $\mathcal{J}\&\mathcal{F}$ score of 83.3.

### 5.4 Ablation Study

Table 4: Ablation study using the validation set of the DAVIS17 benchmark [28].

| Variants | $\mathcal{J}$ M | $\mathcal{J}$ R | $\mathcal{J}$ D | $\mathcal{F}$ M | $\mathcal{F}$ R | $\mathcal{F}$ D | $\mathcal{J}\&\mathcal{F}$ M |
|---|---|---|---|---|---|---|---|
| Keep first frame +URR | 63.3 | 70.2 | 31.4 | 69.0 | 77.8 | 30.2 | 66.2 |
| Keep latest frame + URR | 66.6 | 76.6 | 19.1 | 70.2 | 83.3 | 21.7 | 68.4 |
| Keep first & latest frames +URR | 71.4 | 82.9 | 17.4 | 74.9 | 85.7 | 21.0 | 73.1 |
| Keep first & latest 5 frames +URR | 69.7 | 79.8 | 19.0 | 73.6 | 84.6 | 20.8 | 71.6 |
| AFB only | 68.5 | 80.4 | 17.3 | 72.0 | 84.0 | 19.5 | 70.2 |
| AFB + Confidence loss $\mathcal{L}_{conf}$ | 71.8 | 83.5 | 15.5 | 73.6 | 84.1 | 19.5 | 72.7 |
| AFB + Local refinement | 70.5 | 80.7 | 17.2 | 73.6 | 85.1 | 22.5 | 72.1 |
| Full (AFB+URR) pretrain only | 59.0 | 66.3 | 27.9 | 62.9 | 70.8 | 31.6 | 60.9 |
| Full (AFB+URR) | **73.0** | **85.3** | **13.8** | **76.1** | **87.0** | **15.5** | **74.6** |

We conducted an ablation analysis of our framework on the DAVIS17 dataset. In Table 4, the quantitative results show the effectiveness of the proposed key modules.

First, we analyzed the impact of the *adaptive feature bank* (AFB), and evaluated 4 memory management schemes, namely, keeping features from (1) the first frame, (2) latest frame, (3) the first and latest frame, and (4) the first and latest 5 frames. The remaining modules follow the full framework (i.e., with uncertain-regions refinement (URR) included). From the first four rows in Table 4, while reference frames help the segmentation, simply adding multiple frames may not further improve the performance. Our adaptive feature bank more effectively organized the key information of all the previous frames. Consequently, our framework AFB+URR has the best performance.

Second, we analyzed the efficiency of the proposed *uncertain-regions refinement* (URR). We disabled URR by training the framework without the confidence loss $\mathcal{L}_{conf}$ in Eqn. 9 or local refinement. Our uncertainty evaluation and local refinement significantly improve performance in these regions, because object boundary regions are often ambiguous, and their uncertainty errors are easily accumulated and can harm segmentation results.

## 6 Conclusion

We presented a novel framework for semi-supervised video object segmentation. Our framework includes an adaptive feature bank (AFB) module and an uncertain-region refinement (URR) module. The adaptive feature bank effectively organizes key features for segmentation. The uncertain-region refinement is designed for refining ambiguous regions. Our approach outperforms the state-of-the-art methods on two large-scale benchmark datasets.

## Broader Impact

Our framework is designed for the semi-supervised video object segmentation task, also known as one-shot video object segmentation. Given the first frame annotation, our model could segment the object of interest in the subsequent frames. Due to the great generalizability of our model, the category of the target object is unrestricted. Our adaptive feature bank and the matching based framework can be modified to benefit other video processing tasks in autonomous driving, robot interaction, and video surveillance monitoring that need to handle long videos and appearance-changing contents. For example, one application is in real-time flood detection/monitoring using surveillance cameras. Flooding constitutes the largest portion of insured losses among all disasters in the world [1]. Nowadays, many cameras in city including traffic monitoring and security surveillance cameras are able to capture time-lapse images and videos. By leveraging our video object segmentation framework, flood can be located from the videos and the water level can be estimated. The societal impact is immersed because such a flood monitoring system can predict and alert a flooding event from rainstorms or hurricanes in time. Our framework is trained and evaluated on large-scale segmentation datasets, we do not leverage biases in the data.

## Acknowledgments and Disclosure of Funding

This work is partly supported by Louisiana Board of Regents ITRS LEQSF(2018-21)-RD-B-03 and National Science Foundation of USA OIA-1946231.

## Footnotes

*Corresponding author. Codes are available at `https://github.com/xmlyqing00/AFB-URR`.

[2]Long videos link: `https://www.kaggle.com/gvclsu/long-videos`

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
