[Supplementary Material]

# Video Object Segmentation with Adaptive Feature Bank and Uncertain-Region Refinement

**Yongqing Liang, Xin Li,** * **Navid Jafari**
Louisiana State University
{ylian16, xinli, njafari}@lsu.edu

**Qin Chen**
Northeastern University
q.chen@northeastern.edu

## 1 Qualitative Comparisons on Long Videos

We show the qualitative comparisons on long videos. Long videos[2] consist of three randomly selected videos from the Internet, that are longer than 1.5K frames and have their main objects continuously appearing. Each video has 20 uniformly sampled frames manually annotated for evaluation.

Figure 1 shows the qualitative results of the long video *dressage* which consists of 3,589 frames. RVOS [5] and A-GAME [1] failed to segment the main object after the 1K frames because they only use the first and the latest frame as cues. STM [3] made some errors on boundary areas because their feature bank can store only a fixed number of key frames, which may miss important information when segmenting long videos. Our adaptive feature bank (AFB) can dynamically manage the large numbers of features, learnt from previous frames, through adaptive merging and removal. With AFB, our algorithm can better handle videos with any length than existing algorithms.

RVOS (CVPR19), $\mathcal{J}\&\mathcal{F}$ score 12.2

A-GAME (CVPR19), $\mathcal{J}\&\mathcal{F}$ score 50.3

STM (CVPR19), $\mathcal{J}\&\mathcal{F}$ score 79.3

Ours (AFB+URR), $\mathcal{J}\&\mathcal{F}$ score 83.3

Figure 1: Qualitative results of the long video *dressage*. We show challenging frames for comparison. Some error regions are marked in blue boxes.

# 2 Qualitative Comparisons on DAVIS17

We show the qualitative comparisons on DAVIS17 dataset [4]. Due to the limited space, we select 6 recent methods, RANet [8], SIAMMASK [7], AGAME [1], FEELVOS [6], RVOS [5] and RGMP [2]. We use their official precomputed segmentations for comparison. Our method (AFB-URR) is trained on static images and the training videos from DAVIS dataset. In Figures 2 to 6, we use green/white bounding boxes to highlight the errors for visualization.

Figure 2: Qualitative results of the test video *drift-chicane* from the DAVIS 17 dataset. We show the segmentation on a challenging frame (frame #49) where the car is surrounded by smoke.

Figure 3: Qualitative results of the test video *bmx-trees* from the DAVIS 17 dataset. We show two challenging frames (#19 and #70) where the boy is occluded.

Figure 4: Qualitative results of the test video *gold-fish* from the DAVIS 17 dataset. 5 fishes in the scene are required to be segmented from the background.

Figure 5: Qualitative results of the test video *India* from the DAVIS 17 dataset. We select three challenging frames (#39, #49, and #72) when the people of interest are occluded by the man who is closer to the camera. The results of the frames #49 and #72 are shown in the next page.

Figure 6: Qualitative results of the test video *India* from the DAVIS 17 dataset. The results of the frames #49 and # 72.

**Failure cases.** In addition, we show and analyze two failure cases of our framework. In Figure 7, our AFB-URR misclassifies the hairs at their boundary. The two hair regions have similar appearance and are connected. In AFB-URR, we encode the input frame into feature maps in low resolution, whose spatial size is $1/16$ of the original size. When the spatial size of the feature map is too small, adjacent objects may refer to the same position in the feature map for segmentation. Consequently, our framework can't distinguish such similar and adjacent objects.

In Figure 8, the appearances of the two dogs are alike and the right dog is misclassified. Similarly, the reason could be that the low-resolution feature map may lack sufficient semantic cues to distinguish two similar and small objects (dogs). By increasing the resolution of the feature maps (using multiple GPU cards), small objects can be better captured and classified. We also plan to explore the spatial information of the target objects to help improve the quality of the segmentation as well.

Figure 7: Qualitative results of the test video *lab-coat* from the DAVIS 17 dataset. These persons are standing together, having similar hairs, and all in white clothes.

## Footnotes

*Corresponding author. Codes are available at `https://github.com/xmlyqing00/AFB-URR`.

[2]Long videos link: `https://www.kaggle.com/gvclsu/long-videos`