[Reviews · NeurIPS 2020]

Review 1

Summary and Contributions: The paper proposes a method for Visual Object Segmentation (multi-objects, the semi-supervised task - first frame is given) that keeps the appearance history of the object in an adaptive way, claiming better organization. The authors base the retrieval from feature bank on the key-value principle. They also use a score for the segmentation uncertainty and penalize it at the loss level. The experimental results are good, having a clear advantage on unseen objects compared with previous SOTA on YouTube-VOS18 and SOTA results on DAVIS2017.

Strengths: The paper is technically sound and the need for an adaptive memory for both the appearance and for the semantics is clearly motivated. The “Local Refinement Mechanism” works like a binarization step, forcing the solution to be closer to the final/wanted 0/1 map. The experiments are done on two datasets, with SOTA results. The solution is novel to a certain degree. Mostly, it brings together well validated principles and ideas from tracking and segmentation and nicely connects them in a derivable pipeline.

Weaknesses: The need for Local Refinement Mechanism is not validated in the ablation. The value of pretraining on the image sets is also not taken into account in the ablation or interpreted in text (how the number of images/classes/datasets influences the score?). The solution is lacking a running time performance analysis, which is quite important in the field.

Correctness: The claims are fair, the method is correct and the empirical validation is convincing.

Clarity: The paper is clearly written and easy to read and follow.

Relation to Prior Work: The prior work references are integrated in the paper right where they are needed, without having a special section. The contributions and the positioning in the fields compared with other methods are clear.

Reproducibility: Yes

Additional Feedback: 1. Final score without Local Refinement Mechanism component 2. Ablation on pretraining 3. Running time comparative analysis Typos: “the the first“, “We minimize our loss uing” Fig. 4 is not that interesting, it is a waste of space. ======================================== After reading the authors' feedback, most of my concerns are addressed: runtime performance analysis, ablation on local refinement component and pretraining. I will keep my rating and recommend acceptance of this paper.


Review 2

Summary and Contributions: In this paper, authors propose a VOS framework, that achieves STOA on Davis17 and on par performance on youtube-VOS. The main contributions of this framework are feature bank, confidence loss, and local fine-grained segmentation module.

Strengths: The proposed model achieves STOA on Davis17 and on par performance on youtube-VOS with STM.

Weaknesses: 1. More experiments should be added for further justification. Table3, author only reported AFB without URR. However, another crucial experiment is AFB with the uncertainty loss but without local fine-grained segmentation module. 2. Further analysis of AFB should be conducted. According to Table3, AFB without URR achieve 70.2 J&F mean and 68.5 J mean. However STM baseline archives 72.2 J&F mean and 69.3 J mean, which makes the story of AFB doesn’t hold. Line 35-37, The author mentions "AFB is more efficient then STM because when the given video becomes longer, STM often occurs miss sampling or encounter out-of-memory crash. " Such claim should be addressed in experiments. 3. Related work about both uncertainty loss and local fine-grained segmentation module should be discussed. Fine-grained segmentation module is not new in 2d image segmentation community. For example[1] and [2]. 4. In supp material, qualitative results comparing to *STM* are missing. STM is the current STOA method which checkpoint released. Qualitative results can't be omit 5. In line 233, author discussed comparison to STM on youtube-VOS unseen class. The terms of "somewhat better" and "lack of computation power" are not valid excuses when compare with STOA method. This is a red sign of lacking systematically analysis. With regard to computational power difference, a fair comparison can be achieved by running STOA method under authors environment. [1] Pointrend: Image segmentation as rendering, Kirillov, Alexander and Wu, Yuxin and He, Kaiming and Girshick, Ross [2] Shapemask: Learning to segment novel objects by refining shape priors, Kuo, Weicheng and Angelova, Anelia and Malik, Jitendra and Lin, Tsung-Yi

Correctness: yes

Clarity: The paper is well written and easy to understand in general.

Relation to Prior Work: No. The paper mainly discussed previous work in Video object segmentation domain. However, related work discussion to 2D image segmentation prior work are totally missing.

Reproducibility: Yes

Additional Feedback: In general, the paper presents impressive results on two benchmark. However, the lack of related-work discussion and the lack of qualitative result comparison to STOA (STM) makes the contribution vague. Furthermore, a more detailed ablation study on URR should be conducted. ########### After retbuttal ########### After reviewing authers' response and discussing with all other reviewers. I believe this paper reaches the accpetance bar. In addition, as authors mentioned, I strongly advice the authors should demonstrate the efficiency in the new long sequence dataset in the next version.


Review 3

Summary and Contributions: In this paper, a new semi-supervised video object segmentation approach is proposed. The paper improves the previous memory matching based VOS approaches (e.g. STM [22]). The authors propose an efficient memory update rule. And, a loss function and module to enhance the boundary accuracy is proposed.

Strengths: - New memory update rule for memory-based VOS algorithm is proposed. In a previous memory based approach [22], the memory is maintained through very simple rule. In this paper, the authors propose a more elegant way to maintain memory that can absorb new features and removes obsolete features. In specific, a threshold is set to distinguish the similarity between memory slots, and LFU index is used to remove useless memories. - Uncertain-Regions refinement (URR). One of the challenges in VOS is computing a sharp boundaries between objects and the background. As shown in Table 3, the proposed URR significantly enhance the accuracy. I wonder if URR module is general so It is applicable to other baseline VOS algorithm.

Weaknesses: - The paper lacks the analysis on time complexity. One of the main contribution is an efficient memory management by absorbing new and removing old features. However, there is no analysis on how the proposed memory management is efficient compared to the naive (or previous) approaches. It would be great if the author provide an extensive analysis on this aspect.

Correctness: The authors' claims and method look correct.

Clarity: The paper is written clearly.

Relation to Prior Work: The method resembles the previous memory-based method (STM [22]) in that they use key-value memory. The authors improve the way to update memory. The relation to STM should be discussed more.

Reproducibility: Yes

Additional Feedback: In Table 1, the numbers for STM is different from the leaderboard at the DAVIS benchmark.


Review 4

Summary and Contributions: The paper proposes a framework for video object segmentation, which consists of 1) a feature bank that dynamically discards or updates features from past frames and their predictions, and 2) a mask refinement module based on the output uncertainty. Authors evaluate their method on the DAVIS and YouTube VOS benchmarks, achieving state of the art performance compared to previous works.

Strengths: - The paper is well written and easy to follow. - The method is well motivated and clearly described. - The paper addresses a relevant and challenging problem in the field and achieves better results with respect to previous works. - Authors report results on 2 different well established VOS datasets and achieve superior performance w.r.t. previous works.

Weaknesses: - The paper is missing a literature review / related work section. While previous works are cited, and authors compare their results w.r.t. them, the paper lacks the context in which the proposed method was built upon. Previous works in the literature (many of which are cited in this paper) have already addressed the problems that this paper aims at solving, namely 1) leveraging information from past frames in the video to make predictions in the current frame, and 2) proposed refinement modules for VOS. Although many of these works are indeed cited, authors do not explicitly mention the relationship between those works and their method, in terms of how they addressed the issues that their approach is trying to solve, and how do their contributions compare to the components of existing approaches designed specifically to address these problems. Although this paper's results are better than those reported in previous works, the scientific contributions are ultimately what matters to the community to build on top of in order to make consistent and grounded progress. - Given the above, I am not fully convinced that this paper's contributions (dynamic feature bank to leverage past information & uncertainty based mask refinement) are having significant impact to the performance that is reported in the paper. There's no empirical evidence suggesting that it's better to use this paper's methodology to refine masks and deal with past frame information instead of those in e.g. [15, 22]. While it is clear that this method achieves better performance on these benchmarks, these comparisons are always performed at the "full model" level, in which we know there can be many other factors involved that have an impact in performance (eg augmentation strategies, choice of hyperparameters, early stopping criteria, etc). For example, one may wonder what would happen if we plug other refinement modules (eg the one from [15]) into the pipeline instead of the one the authors propose? Is *this* refinement method better than others? There are no experiments that suggest so. - No runtime numbers are reported. I would have expected authors to include an analysis on that respect. The proposed architecture uses two separate ResNet based encoders: one for the query and one for previous frames and their predictions. Further, authors mention that the context encoder is used separately on each individual object that needs to be predicted, so I wonder about the complexity of the full pipeline, and how does this method compare to existing ones. Recent works eg [15, 22] report runtime numbers, so I would have expected authors to do the same here.

Correctness: To the best of my knowledge, the proposed method is correctly and fairly evaluated and compared to existing works. The method is clearly described and is scientifically sound.

Clarity: Yes, the paper is well written and easy to follow.

Relation to Prior Work: As previously mentioned, the paper is missing a literature review section. Although previous works are mentioned in the introduction, authors should explicitly address the differences between their approach and existing ones. Further, there are some recent works that are not cited in the main manuscript, although authors qualitatively compare their results to them in the supplementary material (references [4] and [6] in the supplementary material). Other than the missing works from the supplementary, there are other related works that are missing: [a] Zheng et al. DMM-Net: Differentiable Mask-Matching Network for Video Object Segmentation. ICCV 2019 [b] Hu et al. Motionguided cascaded refinement network for video object segmentation CVPR 2019

Reproducibility: Yes

Additional Feedback: - Table 1 goes beyond the paper margins. I suggest that authors rearrange it and make some tweaks to make it fit. - check typos in work laptop - L89 closet value --> closest value - Please provide dimensions of the variables that are used in equation 2, and double check the definition of the loss. If I am not mistaken, the model is optimized with pixel-level cross entropy loss, where each pixel is expected to belong to one of the classes (video instances + background). So the loss for pixel j is defined by: L_j = - sum_{i} t_i_j*log(f_i_j), for i=1:C, where f_i_j is the score given by the model for class i at pixel i, and t_i_j is the ground truth value for class i and pixel j. Then the total loss should be the average of the loss for all pixels. -- After authors' feedback: Authors addressed most of my questions regarding the relation of their contributions wrt previous work, and runtime performance analysis. Reported FPS numbers in the rebuttal are significantly better than previous works, which speaks in favor of their method. I encourage authors to add this analysis in the revision. Further, I strongly suggest authors to include the discussion about key differences of their work wrt previous works (which they provided as an answer to Q1 in the rebuttal) in the final version.

[Author Response · NeurIPS 2020]

We thank reviewers for their comments. Our responses to reviewers' (R1-R4) comments are itemized as follows.

**Q1. Key difference of (i) Adaptive Feature Bank (AFB) vs STM (R3), (ii) AFB vs other VOS work (R1, R4).**

**(i)** 1) Our AFB is the first module that does not uniformly sample frames but dynamically manage object's key features
in VOS literature. 2) STM uniformly stores every one of $K = 5$ frames in feature bank, which will fill up 1080Ti
MEM (11GB) when a single-object video has 350+ frames. Practical videos are often longer (e.g., avg. YouTube
video has $\approx 12$ minutes or $22K$ frames). To process a 10-min video, STM needs to set $K = 300$ and misses many
important frames (see **Q4**). In contrast, AFB performs dynamic feature merging and removal, and can handle videos
with any length effectively. **(ii)** Most recent methods (other than STM) only store features from the first and/or last
frames. Specifically, R4 mentioned four papers: DMM-Net [a] stores the 1st frame, FEELVOS [4] and RANet [6] store
the first and latest frame. Motion-guided [b] is not a matching-based, but is in another category, mask-propagation
based methods. Mask propagation methods are unstable when objects undergo significant occlusions. [4] and [6] were
compared in Table 1. [a] has a $J\&F$ score of 70.7 on DAVIS17 (lower than STM 71.6, and ours 74.5). [b] reports $J$
76.4 and $F$ 75.7 in its Table 4 on DAVIS16, while our scores are $J$ 85.5 and $F$ 83.4.

**Q2. Key difference of Uncertainty Region Refinement (URR) vs existing work (R2, R4).**

(R2) Thank you. We will discuss fine-grained segmentation work in **image segmentation** (e.g. *PointRend* [1] and
*ShapeMask* [2]) in the revision. But our URR is the first module addresses boundary refinement in VOS literature.
Meanwhile, the refinements in [1] and [2] are different from ours. *ShapeMask* [1] revised the decoder from FCN, and
such global decoders can fail to recover accurate masks on boundaries. Our URR has a strong local regularization on
uncertain regions to predict precise mask on boundary. *PointRend* [2] was published in CVPR2020, after this paper's
submission. Its differences from ours are: **(i)** *PointRend* defines uncertainty on a binary mask, while ours is defined
on more general multi-object classification (score ratio of the top-1 class to top-2 class). **(ii)** Furthermore, *PointRend*
does a one-pass detection and refinement on uncertain regions. While we design an uncertainty loss to generate cleaner
object masks (see Fig. 3). As R1 pointed out, this works like a binarization step, forcing the solution to be closer to
the final/wanted 0/1 map. **(iii)** For refinement, *PointRend* refines uncertain points by their own features. While we use
features from reliable points to refine the uncertain ones, through a non-local mechanism.

URR vs refinement module in [15, 22] (R4): Actually, the refinement module [15, 22] is used in our decoder to generate
initial masks (see Line 97), which are inaccurate on boundary. Then, URR effectively refines these boundary regions.

**Q3. Runtime performance analysis (R1, R3, R4)**: Our model has better runtime performance than the baseline STM.
On DAVIS17, with an NV. 1080Ti, STM achieves 3.4fps ($J\&F$ 71.6), and ours achieves 3.9fps ($J\&F$ 74.5). Our
runtime is a trade-off between latency and accuracy depends on requirement: if we limit the memory usage under $20\%$,
it achieves 5.7fps ($J\&F$ 71.7). We will add detailed runtime comparison with other SOTA to the manuscript.

**Q4. Why is AFB important? Is AFB w/o URR worse than baseline STM? (R2).**

**(i)** AFB optimizes features management, leading to less memory usage (see **Q1**) and less runtime latency (see **Q3**)
compared with STM. **(ii)** As Table 3 shows, compared with STM, AFB w/o URR has slightly downgraded $J\&F$ (from
72.2 to 70.2), but greatly reduces ($\sim 75\%$) memory usage. Current benchmarks YouTube-VOS (132 frames avg.) and
DAVIS17 (67 frames avg.) only contain short clips, which cannot show the advantages of AFB in dealing with long
videos in real-world (see **Q1**). **(iii)** A new dataset of long videos (2K+ frames each) is added to show the performance
in real-world, on which AFB w/o URR vs STM are 82.9 vs 66.5 in $J\&F$ score. This dataset will be released with paper.

**Q5. Effectiveness of two sub-components of URR: uncertainty loss (R2), and local refinement mechanism (R1).**

We performed these two new experiments (b, c) on DAVIS17. The $J\&F$ scores are: (a) AFB only: 70.2; (b) AFB +
uncertainty loss: 72.7; (c) AFB + fine-grained module: 72.1; (d) AFB + URR (full model): 74.5.

**Q6. On DAVIS17: Qualitative comparison with STM (R2)? STM has better scores in Leaderboard (R3)?**

(R2) We followed most SOTA settings that train models only using DAVIS17. The released STM checkpoint was
trained on DAVIS17+YouTubeVOS. Also, STM authors didn't release training codes or corresponding results for
qualitative comparison. (R3) The STM scores in leaderboard was also trained on DAVIS17+YouTubeVOS, we used the
right score that only uses DAVIS17 from its paper.

**Q7. Red Sign: On YouTubeVOS unseen, lack comparison by running SOTA (STM) in same environment (R2).**

We can only compare numbers, not re-run STM codes because (a) STM didn't release training codes (see STM GitHub
*Reproducibility* Issue #6); (b) STM didn't plan to release model for YouTubeVOS comparison (GitHub Issue #3).

**Q8. Performance for only pretraining on image sets in ablation study (R1)**: On DAVIS17, only pretrained by
image datasets: $J\&F = 60.9$; after training on DAVIS17, $J\&F = 74.5$.

[Meta-Review · NeurIPS 2020]

The paper presents a new approach for Visual Object Segmentation that keeps the appearance history of the object in an adaptive way, New memory update rule for memory-based VOS algorithm is proposed. New memory update rule for memory-based VOS algorithm is proposed. The method is well motivated and clearly described. There are some important literature missing and the analysis is not complete , e.g. in term of time complexity. All reviewers discussed for several concerns , that have been partially overcame by the rebuttal. After Rebuttal reviewers agree on the fact that the paper is Marginally above the acceptance threshold.